# Effect of the Addition of Lemongrass (*Cymbopogon citratus*) on the Quality and Microbiological Stability of Craft Wheat Beers

**DOI:** 10.3390/molecules27249040

**Published:** 2022-12-18

**Authors:** Justyna Belcar, Józef Gorzelany

**Affiliations:** 1Department of Food and Agriculture Production Engineering, University of Rzeszów, 4 Zelwerowicza Street, 35-601 Rzeszów, Poland; 2Doctoral School of the University of Rzeszów, University of Rzeszów, st Rejtana 16C, 35-959 Rzeszów, Poland

**Keywords:** lemongrass, wheat beer, beer quality, antioxidant potential, microbiological stability of beers

## Abstract

Lemongrass (*Cymbopogon citratus*) is a valuable source of vitamins, macro- and microelements, and essential oils. The purpose of this study was to compare the physicochemical properties, sensory properties, antioxidant activity, and microbiological stability of wheat beers enriched with varying additions of crushed lemongrass. Sensory evaluation showed that wheat beers enriched with 2.5% *m*/*v* lemongrass were characterised by balanced taste and aroma (overall impression). Physicochemical analysis of the wheat beers showed that increasing the concentration of lemongrass in the finished product negatively affected the ethanol content. Alcohol content in wheat beer enriched with 1% *m*/*v* lemongrass was on average 14.74% higher than wheat beer enriched with 2.5% *m*/*v* lemongrass and on average 17.93% higher than wheat beer enriched with 5% *m*/*v* addition of lemongrass. The concentration of lemongrass in the finished product also increased the acidity of the beers and affected the colour of the wheat beers compared to the control beer. The total polyphenol content and antioxidant activity of lemongrass-enriched wheat beers varied. Of the lemongrass-enriched beers analysed, the beer product with 5% *m*/*v* lemongrass was the most microbiologically stable. According to the study, crushed lemongrass-enriched wheat beer may represent a new trend in the brewing industry, but the brewing process still needs to be improved.

## 1. Introduction

In recent years, consumer interest in craft beers produced by craft breweries and microbreweries has been growing. Craft beer beverages provides the opportunity to produce new, unprecedented, and often amazing combinations of plant-based raw materials with beer which affects the flavour and aroma qualities [1]. Among the various beer styles produced by craft breweries are wheat beers. Wheat beers use wheat malt or unmalted wheat grain to replace part of barley malt (generally 40 to 60% of the hopping) [2,3]. Wheat beers are characterised by an intense haze, a fine but stable head, a slight bitterness, and a slightly sweet aftertaste due to the fermentation process carried out, during which compounds are produced that give flavour to the finished product (including phenols, aldehydes, esters and their derivatives) [4,5,6]. Wheat beers contain many health-promoting substances in their composition, including polyphenols, micronutrients, vitamins (folate, riboflavin, pantothenic acid, pyridoxine, and niacin), fibre, antioxidants at a relatively low ethanol content (usually 4.0–6.0% *v*/*v*) [7,8,9,10]. Antioxidants protect the human body from oxidative stress, and are very sensitive to temperature, pH, oxygen levels, light or yeast load. During the ageing of beer, they are responsible for flavour changes [9]. Antioxidant compounds, through their action, trap free radicals from the body, which are targeted to attack and damage the structure of DNA, membrane lipids, or proteins [2,11].

Herbs or their extracts are increasingly being added to beers to give them the right flavour, bitterness, and aroma. The primary raw material used in brewing is hops, but in *gruit*-style beers this is being replaced by yarrow, marshmallow, or European waxwort, although less common, spice additions such as juniper berries, caraway seeds, or aniseed. In recent years, new flavour combinations have been sought in which herbal notes play a central role, including lavender, ginger, coriander, heather, cardamom, nettle, liquorice, angelica, lemongrass, sage, dandelion or nettle [1,7,12]. Each of the herbs added to beer through their effects (e.g., bactericidal and fungicidal, sedative, antidiabetic, or antioxidant, depending on the species of plant used) improve the health-promoting properties of the finished beer product, but can also have adverse effects when consumed in excessive amounts, for example, cause psychoactive or abortifacient effects, for example, juniper [7,12].

Lemongrass is a perennial plant, grown mainly in countries with tropical climates (Asia, South America, and Africa), and is characterized by a distinctive lemony aftertaste, giving its products a positive taste. Lemongrass is known to have a wide range of health-promoting properties, including that it can be used as an anti-inflammatory, analgesic, or antiseptic agent [13,14,15]. Lemongrass contains phenolic compounds, vitamins; thiamine, niacin, pantothenic acid, riboflavin, ascorbic acid or folic acid; macro and micronutrients: Mg, P, K, Zn, Cu, Fe [13]. Lemongrass also contains essential oils (on average 1–2% in dry matter), including citral (70–80% of the total essential oil content of lemongrass), that show potential antimicrobial activity, which is important in beer production from the point of the view of microbiological purity of the finished product, but also antidiabetic, diuretic, or anticancer [13,14,15]. Lemongrass is widely used as both fresh raw material or can be subjected to a drying process and in powder form used in gastronomy (Asian dishes), but also in the food industry for flavouring teas, as an additive to meats, sauces, wines, dairy desserts, beverages, sweets, and used as a flavouring or fragrance used in bakery and confectionery [13,14,16].

The purpose of this study was to compare the physicochemical properties, sensory properties, antioxidant activity, and microbiological stability of wheat beers enriched with varying concentrations of crushed lemongrass and to assess the practical applicability of the findings to expand the range of wheat beers.

## 2. Results and Discussion

### 2.1. Physicochemical Characteristics of Wheat Beers

The results of the evaluation of the physicochemical parameters of the wheat beers enriched with crushed lemongrass are shown in Table 1.

Wheat beers enriched with lemongrass were characterised by a significantly lower apparent extract (on average 35.04% compared to the control, that is, beer marked CB), moreover, wheat beers enriched with the addition of lemongrass of 1% *m*/*v* and 2.5% *m*/*v* were characterised by a significantly lower real extract of 1.85% *m*/*m* (BL1) and 2.31% *m*/*m* (BL2; Table 1), respectively. The lowest value of the original extract was characterised by the beer BL2 (10.67% *m*/*m*), while the highest value was obtained for the control beer (CB; Table 1).

The fermentation of wheat beers affects not only the taste and aroma qualities of the finished product, but also an important component such as the ethanol content [17]. Of the wheat beers analysed, the highest degree of final apparent attenuation was characterised by beer enriched with 5% *m*/*v* lemongrass addition (BL3), while the highest degree of final real attenuation was obtained for BL1 beer (84.05%; Table 1). Furthermore, increasing lemongrass addition to wheat beers resulted in a significant decrease in the degree of real attenuation, which was simultaneously correlated with the ethanol content of the finished product. The highest ethanol content was characterized by CB beer (5.94% *v*/*v*), while among lemongrass-enriched beer products, its 1% *m*/*v* addition had a positive effect on the content of the parameter in question and was higher on average by 14.74% compared to BL2 beer and by 17.93% on average compared to BL3 beer (Table 1). According to Tomova et al. [18], the addition of various herbs to beers, including thyme, oregano, cinnamon, or cloves, causes the essential oils they contain to inhibit the production of ethanol by yeast. In a study by Nordini and Garaguso [19], orange peel-enriched beers had an ethanol content of 6.0% *v*/*v*. In a study by Baigts-Allende et al. [20], citrus-enriched beers were characterised by an alcohol content of 4.0–8.2% *v*/*v*, while in the study of Patraşcu et al. [21], the ethanol content of lemon beers ranged from 1.9 to 4.0% *v*/*v* and that of grapefruit beers from 1.9 to 2.5% *v*/*v*. The caloric content of the wheat beers varied; the control beer (CB) had the highest value, mainly due to its ethanol content, while the lemongrass beers had a caloric content of 39.28–45.42 kcal/100 mL and were on average 22.61% lower than the control (CB; Table 1).

The colour of the beers depends significantly on the malt composition used for mashing. The addition of lemongrass affected the colour of wheat beers, and an increase in the concentration of lemongrass in the finished product significantly affected the colour of the beers, which ranged from 16.9 to 21.5 EBC units (Table 1, Figure 1). In a study by Baigts-Allende et al. [20], citrus-infused barley beers were characterised by a colour level of 5.8 EBC units. Patraşcu et al. [21] evaluated the colour in lemon and grapefruit beers and it was at 6.75–6.83 EBC units and 16.98–17.36 EBC units, respectively.

With increasing lemongrass addition to wheat beers, there was a decrease in its pH on average of 3.76% compared to the control (CB), while the acidity of lemongrass enriched beers increased significantly (by an average of 20.71%, including the highest acidity of beers with 5% *m*/*v* addition (BL3)–5.38. In a study by Nordini and Garaguso [19], orange peel had a pH of 4.86. Patraşcu et al. [21] analysed lemon and grapefruit beers, whose acidity was, respectively: 4.0–4.64 and 4.0–4.4, while the pH of the analysed beers was at levels of: 2.85–3.09 and 3.27–3.49. Beers characterised by relatively low pH are more microbiologically stable due to the reduced growth of undesirable microflora in the finished beer product [22]. 

The carbon dioxide content of the wheat beers was at a similar level (0.43–0.49%; Table 1). In the study by Patraşcu et al. [21], the carbon dioxide content of lemon beers was at the level of 0.48–0.55% and that of grapefruit beers at 0.52%. The taste of bitterness in wheat beers enriched with lemongrass was significantly higher than the control beer (CB) and ranged from 16.1 to 18.5 IBU (Table 1), moreover, the bitterness content increased with increasing lemongrass concentration in wheat beers. The feeling of bitterness in the analysed beers come not only from the basic raw material used to make beers (hops), the degree of α-acid isomerisation during boiling of the wort with hops, and the degree of protein reaction of the proteins with the polyphenols contained in the malt, but also from the additives used [4,23].

### 2.2. Content of Bioactive Compounds in Wheat Beers Enriched with Lemongrass

Polyphenols are chemical compounds that pass during the mashing process from malt (70–80%) and during the boiling process with hops (20–30%) to the finished beer product [24]. Preparation of raw materials, e.g., the degree of fineness, but mainly the mashing and boiling process with hops, significantly determines the content of total polyphenols and their degree of isomerisation in the finished product [23]. Polyphenols are chemical compounds that differ in their chemical structure, which influences their antioxidant activity (including their differential bioactive activity; [25]) and are responsible for the sensory impressions felt by consumers, including the sensation of contentiness, bitterness, acidity, or the sensation of fullness of taste. The content of total polyphenols in lemongrass enriched wheat beers varied and ranged from 182.0 to 264.7 mg GAE/L (Table 2). Wheat beers with 2.5% *m*/*v* of lemongrass addition had the highest total polyphenols, on average 6.23% higher than the control beer (CB; Table 2). In the study by Nardini and Garaguso [19], beers with added orange peel were characterised by a total polyphenol content of 639 mg GAE/L. Commercial Portuguese lemon-flavoured fruit beers were characterised by a total polyphenol content of 240–304 mg GAE/L [26]. Beers enriched with herbs were characterised by a significantly higher total polyphenol content; from 316.67 mg GAE/L for beers enriched with hop cones to 384.22 mg GAE/L for beers enriched with thyme [27]. Beers obtained by Ulloa et al. [28] enriched with propolis were characterized by total polyphenol content ranging from 253–306 mg GAE/L depending on its addition, and beers enriched with *Parastrephia lucida* leaves were characterised by a total polyphenol content ranging from 480–800 mg GAE/L depending on the concentration of the leaves in the beer [29]. The addition of green tea to the beers also significantly influenced the content of total polyphenols in the finished beer product (600 mg GAE/L) [30].

The content of biologically active compounds such as vitamins, bitter and polyphenolic compounds, or melanoidins in beers influences their antioxidant potential [31,32]. Wheat beers generally have a higher antioxidant potential compared to barley beers (depending on the style of beer produced). The addition of lemongrass to wheat beers significantly reduced the antioxidant activity of beers determined by three methods (DPPH^.^, FRAP, ABTS ^+^), regardless of the concentration used, in that an increase in lemongrass content in wheat beers (reduced) the antioxidant activity of the finished product (Table 2). Citral is responsible for the high antioxidant activity (free radical scavenging) of lemongrass, while the origin of lemongrass and the way it is stored and processed significantly determine the antioxidant potential of the raw material [13]. In a study by Nordini and Garaguso [19], orange peel had an antioxidant activity determined by the ABTS method of 2.67 mM TE/L, while the reduction capacity determined by the FRAP method was 5.65 mM Fe^2+^/L. Commercial Portuguese lemon-flavoured fruit beers had antioxidant activity determined by the DPPH method of 0.035–0.037 mM TE/L, while the ABTS method was at 0.008 mM TE/L [26]. Beers enriched with herbs had significantly higher antioxidant activity; from 2.83 mM TE/L for beer enriched with hop cones to 3.72 mM TE/L for beer enriched with thyme (determined by the DPPH method) and from 4.25 mM TE/L for beer enriched with nettle to 4.71 mM TE/L for beer enriched with thyme (determined by the FRAP method; [27]). The beers obtained by Ulloa et al. [28] enriched with propolis were characterised by antioxidant potential depending on their addition; 0.49–0.57 mM TE/L (DPPH method), 1.55–1.89 mM TE/L (FRAP method) and 0.68–0.80 mM TE/L (ABTS method), while enrichment with *Parastrephia lucida* leaves characterised beers with antioxidant activity of 2.17–5.46 mM TE/L (FRAP method) and 1.38–3.34 mM TE/L (ABTS method) depending on the concentration of leaves in the beer [29].

### 2.3. Microbiological Stability of Wheat Beers

Two chemical compounds, α- and β-citral, are responsible for the antibacterial properties in lemongrass, inhibiting the proliferation of Gram(+) and Gram(−) bacteria; in addition, they also have antifungal activity, including against strains of *Fusarium* spp., which are a common pathogen of cereals, including wheat and barley, which produce malts used in the brewing industry [13,15]. Lemongrass has been used successfully as a natural preservative in the storage of fruits, juices, baked goods or dairy products [14]. Table 3 shows the results of the microbiological analysis of lemongrass enriched wheat beers.

Beer is considered a product of relative microbiological stability as a result of the chemical composition of the beverage. After 15 days of bottling, the wheat beers analyzed were characterised by varying yeast and mould counts; from 9.3 × 10^4^ cfu mL^−1^ for wheat beer without lemongrass (CB) to 1.8 × 10^6^ cfu mL^−1^ for beer enriched with 2.5% *m*/*v* lemongrass (Table 3). After 45 days of bottle fermentation of the beers, the number of yeasts and moulds in the finished product significantly decreased by an average of 4 log, with the largest decrease observed for BL3 beer (Table 3). There should be no yeasts in the wort other than those introduced intentionally (e.g., *Saccharomyces cerevisae*) to guide the fermentation process. The presence of other wild yeasts influences the development of an unpleasant phenolic aftertaste in the finished beer product, as well as affects the turbidity of the beer and its excessive carbonation [33]. The highest number of aerobic mesophilic bacteria was determined after 15 days of bottle fermentation of wheat beers for BL1 beer (1.1 × 10^6^ cfu mL^−1^, Table 3). The number of mesophilic lactic fermentation bacteria in the analysed wheat beers independently of the addition of lemongrass was at a very low level. Lactic fermentation bacteria, mainly of the genera *Lactobacillus* and *Pediococcus*, are most often responsible for the infection and spoilage of beers, due to their resistance to the temperatures of the mashing and boiling process with hops from wort and the high ethanol concentration in the finished product [34]. After one month of storage, unpasteurised beers had an average lactic acid bacteria count of 5.83 log_10_ cfu ml^−1^, a yeast count of 4.02 log_10_ cfu mL^−1^, and a residual bacteria count of 1.3 log_10_ cfu mL^−1^ in the finished product. A further month of storage resulted in a slight reduction in the microbial load of the beer [34]. In a fruit beer from a small brewery immediately after bottling, the yeast content was 4.6 × 10^4^ cfu mL^−1^ [35]. Craft beers are most often not pasteurized and thus contain live microorganisms that process the chemical compounds in beer, and autolysis also occurs, so the consumption time of the finished product should not be excessively long [36].

### 2.4. Sensory Analysis of Wheat Beers

Both the appeal and the consumer acceptance of a particular type of beer is an important quality differentiator of the finished product and influences its sales. The taste and aroma qualities of wheat beer enriched with herbs, including lemongrass, may influence consumers’ preference to purchase a particular beer, or this purchase may only be a one-off. The results of sensory evaluation of the fruity wheat beers by the 11-member panel are shown in Table 4 and Figure 2.

Wheat beers enriched with lemongrass were characterised by a significantly more desirable aroma compared to the control (CB); moreover, wheat beers with 2.5% *m*/*v* lemongrass addition were rated best in terms of the taste of the finished product (Table 4). The taste and aroma qualities of the lemongrass-enriched wheat beers were determined by the cereal and malt notes coming from the wheat and barley malts used for the mashing, as well as the intensity and fullness of flavour, freshness, and refreshment, which came from the herbal additive that lemongrass constituted (Figure 2). The taste and aroma of the beer are influenced not only by the raw materials used but also by the products of the fermentation process (such as aldehydes, phenols, or esters) that affect the flavour profile of a particular beer. The stability and persistence of the beer head increased with increasing lemongrass steepness in wheat beer; moreover, the head was characterised by a fine bubbled and creamy structure. Tomova et al. [18] made a different observation for fermented worts with tangerine and grapefruit oil; in the finished product, the amount of carbon dioxide released was negligible. The bitterness sensation in the wheat beers analysed was at a similar level, with the lowest sensation observed for BL1 beer (Table 4). The most attractive beer in terms of carbon dioxide saturation of wheat beer with carbon dioxide and overall sensation for the sensory panel turned out to be the beer enriched with 2.5% *m*/*v* of lemongrass addition (Table 4). The addition of herbs in the production of beers is commonly used in brewing not only for its functional properties, but also for the sensory profile of the finished beer product [37]. In a study by Tomova et al. [18], the addition of tangerine oil to fermenting wort had a positive effect on the consumer acceptability of the finished product. In a study by Dordević et al. [27], the addition of lemon balm extract to beer had a positive effect on the consumer acceptability and sensory qualities of the finished product. Of the herbal additives (dandelion, nettle and sage), the dandelion enrichment of beer influenced the higher palatability and overall acceptability and was preferred by female consumers, while beer enriched with sage was more preferred by men due to the pronounced bitterness of the finished beer product [1].

## 3. Materials and Methods

### 3.1. Material

Winter common wheat of the ‘Lawina’ variety were used for wheat beers from a field experiment conducted in 2021 in Kosina (50°04′17′′ N 22°19′46′′ E), Podkarpackie Province (Poland). A 5-day malt was prepared from wheat grain (the malting process methodology is described in Belcar et al. [38]). The wheat malt (‘Lawina’ variety) had the following characteristics: extract potential—84.1% d.m. (d.m.—dry matter), total protein content—11.5% d.m., content of soluble protein—4.33% d.m., diastatic power—356 WK, and degree of final real attenuation—79.7%. 

The raw material charge for wheat beer brewing consisted of 60% commercial barley malt and 40% wheat malt. Barley malt from the Viking Malt malting plant in Strzegom (Poland) was used in the study. The barley malt had the following characteristics: extraction potential—80.0% d.m., total protein content—11.4% d.m., content of soluble protein—3.75% d.m., diastatic power—324 WK, and degree of real final attenuation—82.1%. Wheat and barley malts were ground to particle size on a FOSS Cemotec disc mill. 

Lemongrass was an addition to wheat beers. Raw material was imported (Fresh World International Co., Bronisze, Poland) and purchased from a local supermarket. The purchased raw material was fresh, it was not processed in any way, it was packed in a vacuum package that was torn open immediately before using the lemongrass. The raw material was crushed by knife manually before being added to the wheat beers.

### 3.2. Production of Beers

The production process was carried out using the infusion method in the laboratory of the Department of Agricultural and Food Production Engineering, University of Rzeszów. Malts weighing a total of 4.0 kg (1.6 kg wheat malt and 2.4 kg barley malt) were mashed and placed in a ROYAL RCBM-40N mash kettle (Expondo; Zielona Góra, Poland; assuming a process efficiency of 80%) and poured over 12.0 L of water (3 L of water for each kilogram of malt). The mashing, boiling process with hops, and cooling of the beer wort were carried out according to the methodology described by Gorzelany et al. [39] and shown on Figure 3. 

Four beer worts were produced, which were characterised by an extract of 12.0 °P (the number of degrees Plato corresponds to the number of kilograms of dry matter (extract) contained in 100 kg of basic wort). After cooling the wort with a coolant, which was water, the worts were transferred to fermentation containers of 30 L each and inoculated with *Saccharomyces cerevisae* Fermentis Safale US-05 yeast (6 × 10^9^/g), which had previously undergone a rehydration process according to the manufacturer’s instructions (0.58 g d.m./L of wort). The fermentation process was carried out for 21 days at 21 °C. After 7 days of fermentation, crushed lemongrass was added to the fermenting wheat beer in a specified amount (1, 2.5, or 5% by volume of wort) and further fermented in fermentation containers. After 21 days, a sucrose solution (0.3%) was added to the beer for refermentation and to obtain the appropriate saturation of the beer and then were bottled. The resulting beers were kept at 20 °C. Sensory and physicochemical tests were performed one month after bottling. 

Wheat beers obtained without and with addition of 1%, 2.5%, and 5% *m*/*v* lemongrass were designated as CB, BL1, BL2, BL3, respectively. 

### 3.3. Analysis of Quality Indicators for Beers

The ethanol content [% *m*/*m* and % *v*/*v*], apparent extract [% *m*/*m*], real extract [% *m*/*m*], and original extract in beer [% *m*/*m*], degree of apparent and real attenuation [%], total acidity [0,1 M NaOH/100 mL], pH, colour [EBC units], carbon dioxide content [%], bitterness content [IBU units] and energy value of beer [kcal/100 mL] were determined according to the methodology described by Belcar et al. [2]. The analyses were performed in three replications.

### 3.4. Total Polyphenol Content and Antioxidant Activity of Wheat Beers

The total polyphenol content [mg GAE/L] using the Folin–Ciocalteu method in the analysed beers was determined according to the methodology described by Gorzelany et al. [39]. The antioxidant activity of wheat beers (using the DPPH [mM TE/L], FRAP [mM Fe^2+^/L] and ABTS [mM TE/L] methods) was determined according to the methodology described by Gorzelany et al. [39]. The analyses were performed in three replications.

### 3.5. Microbiological Stability of Wheat Beers

The number of aerobic mesophilic bacteria was determined in accordance with PB-77/LM issue 5 of 10.02.2022 [40], the number of mesophilic lactic fermentation bacteria was determined according to PN-ISO 15214:2002 [41] and the number of yeasts and moulds was determined in accordance with PN-ISO 21527-1:2009 [42]. Microbiological cultures of the beers were performed twice: after 15 and 45 days after the bottling of the wheat beers.

### 3.6. Sensory Analysis

Sensory analysis was carried out by an expert team of 11 people (seven women and four men aged 30–45 years) in the sensory evaluation laboratory according to the EBC method 13.13 [43]. Beer samples were served after cooling to 10 °C coded in random order in 250 mL of transparent plastic cups. Oral water was administered between each evaluation. Sensory analysis of the beers was performed according to the methodology described in Belcar and Gorzelany [44].

### 3.7. Statistical Analysis

The results of the wheat beer analyses are presented as mean values with standard deviation (*n* = 3). Statistical analysis of the results was performed using Statistica 13.3 statistical software (TIBCO Software Inc., Tulsa, OK, USA). The ANOVA analysis of variance was used in the analyses with a significance level of α = 0.05. Comparisons of mean values were made using the HSD-Tukey test.

## 4. Conclusions

Lemongrass at a concentration of 2.5% *m*/*v* added in wheat beer showed the most balanced flavour profile. Additionally, this beer was characterized by a higher total poly-phenol content average about 6.23% and lower calorie content average about 28.36% compared to the untreated control wheat beer. The finished beer product enriched with 1% *m*/*v* of lemongrass addition was characterised by good physicochemical properties and relatively high antioxidant activity (for example DPPH test—2.46 mM TE/L), while the highest concentration of lemongrass in wheat beer (5% *m*/*v*) had a positive effect on the microbiological stability of the finished product. The enrichment of wheat beers with lemongrass may represent a new direction to expand the range of wheat beers; however, it still requires technological improvements. 

## Figures and Tables

**Figure 1 molecules-27-09040-f001:**
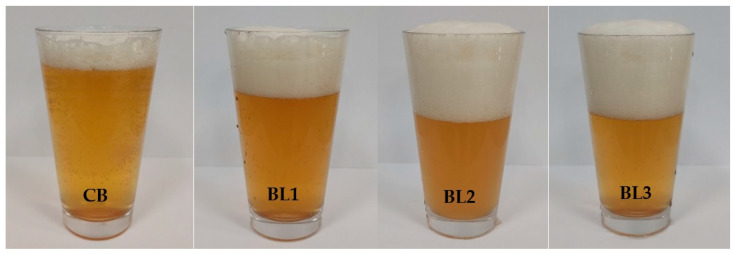
Appearance of the wheat beers obtained from left: control (CB), wheat beer with 1% *m*/*v* (BL1), 2.5% *m*/*v* (BL2) and 5% *m*/*v* (BL3) lemongrass addition.

**Figure 2 molecules-27-09040-f002:**
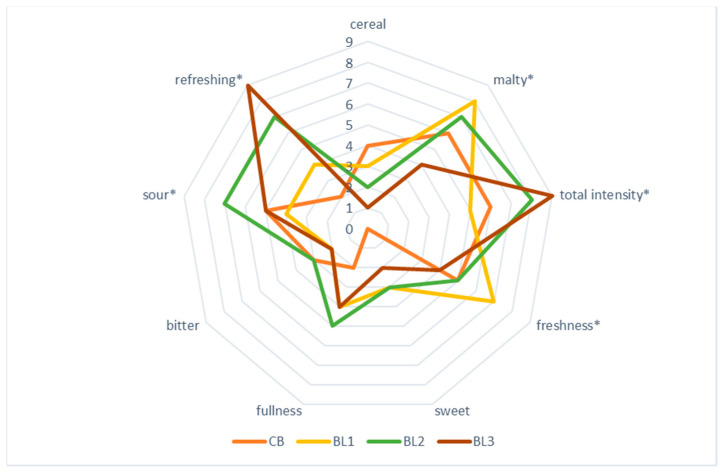
Sensory profile of wheat beers control (CB) and with 1% *m*/*v* (BL1), 2.5% *m*/*v* (BL2) and with 5% *m*/*v* (BL3) lemongrass addition. (* indicates attributes which were statistically different at *p* < 0.05).

**Figure 3 molecules-27-09040-f003:**
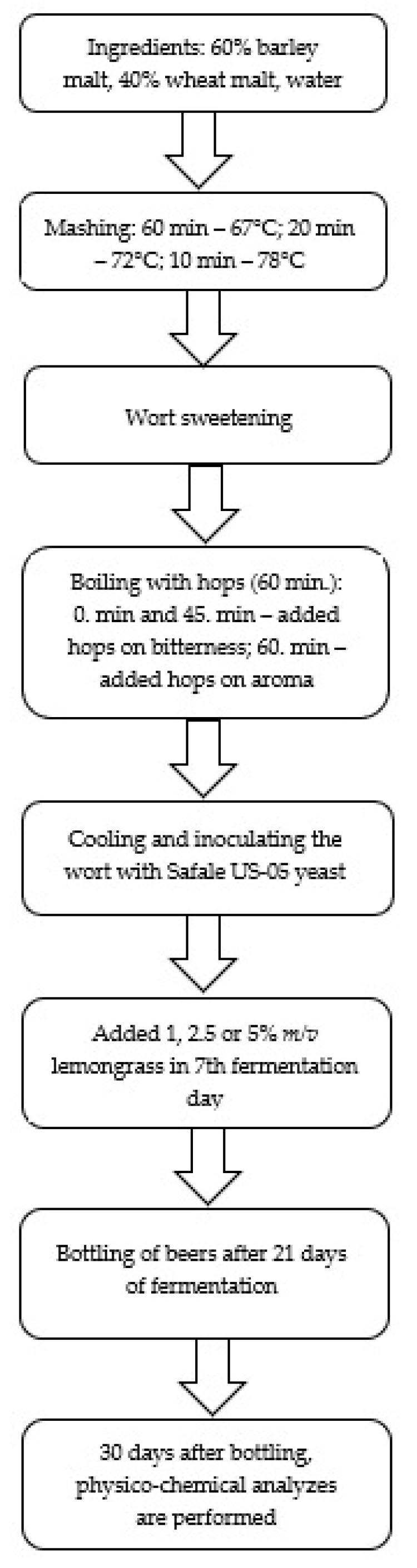
Production process of wheat beer enriched with lemongrass.

**Table 1 molecules-27-09040-t001:** Results of the physicochemical analysis of wheat beers with lemongrass added.

	CB	BL1	BL2	BL3
Apparent extract [%; *m*/*m*]	3.52 ^c^ ± 0.06	2.58 ^b^ ± 0.08	2.06 ^a^ ± 0.06	2.22 ^a^ ± 0.22
Real extract [%; *m*/*m*]	3.27 ^c^ ± 0.07	1.85 ^a^ ± 0.05	2.31 ^b^ ± 0.01	4.08 ^d^ ± 0.08
Original extract [%; *m*/*m*]	14.62 ^d^ ± 0.08	11.60 ^b^ ± 0.10	10.67 ^a^ ± 0.06	12.03 ^c^ ± 0.03
Degree of final apparent attenuation [%]	75.92 ^a^ ± 0.10	77.76 ^b^ ± 0.06	80.69 ^c^ ± 0.09	81.55 ^d^ ± 0.05
Degree of final real attenuation [%]	77.63 ^b^ ± 0.04	84.05 ^d^ ± 0.05	78.35 ^c^ ± 0.04	66.08 ^a^ ± 0.08
Content of alcohol [%; *m*/*m*]	5.94 ^d^ ± 0.05	5.02 ^c^ ± 0.05	4.28 ^b^ ± 0.08	4.12 ^a^ ± 0.08
Content of alcohol [%; *v*/*v*]	4.73 ^c^ ± 0.05	4.00 ^b^ ± 0.10	3.40 ^a^ ± 0.05	3.28 ^a^ ± 0.07
Colour [EBC units]	22.7 ^d^ ± 0.6	21.5 ^c^ ± 0.5	19.3 ^b^ ± 0.2	16.9 ^a^ ± 0.7
Titratable acidity [0.1 M NaOH/100 mL]	3.82 ^a^ ± 0.05	4.36 ^b^ ± 0.06	4.82 ^c^ ± 0.02	5.38 ^d^ ± 0.08
pH	4.79 ^c^ ± 0.04	4.68 ^b^ ± 0.03	4.63 ^b^ ± 0.03	4.52 ^a^ ± 0.02
Bitter substances [IBU]	14.4 ^a^ ± 0.3	16.1 ^b^ ± 0.1	17.7 ^c^ ± 0.4	18.5 ^d^ ± 0.5
Content of carbon dioxide [%]	0.43 ^a^ ± 0.02	0.47 ^a^ ± 0.07	0.49 ^a^ ± 0.04	0.47 ^a^ ± 0.03
Energy value [kcal/100 mL]	54.83 ^d^ ± 0.07	42.59 ^b^ ± 0.07	39.28 ^a^ ± 0.04	45.42 ^c^ ± 0.10

Data are expressed as mean values (*n* = 3) ± SD; SD—standard deviation. Mean values within rows with different letters are significantly different (*p* < 0.05). CB—wheat beer without lemongrass added; BL1—wheat beer with 1% *m*/*v* lemongrass added; BL2—wheat beer with 2.5% *m*/*v* lemongrass added; BL3—wheat beer with 5% *m*/*v* lemongrass added.

**Table 2 molecules-27-09040-t002:** Total polyphenol content and antioxidant activity of wheat beers.

	CB	BL1	BL2	BL3
Total polyphenol content [mg GAE/L]	248.2 ^c^ ± 0.5	230.8 ^b^ ± 0.8	264.7 ^d^ ± 0.06	182.0 ^a^ ± 0.5
DPPH^.^ [mM TE/L]	2.38 ^c^ ± 0.08	2.46 ^c^ ± 0.06	1.70 ^b^ ± 0.05	1.08 ^a^ ± 0.08
FRAP [mM Fe^2+^/L]	2.42 ^c^ ± 0.08	1.56 ^b^ ± 0.10	0.92 ^a^ ± 0.08	1.62 ^b^ ± 0.07
ABTS^+·^ [mM TE/L]	0.92 ^b^ ± 0.10	1.46 ^d^ ± 0.06	0.79 ^c^ ± 0.01	0.46 ^a^ ± 0.06

Data are expressed as mean values (*n* = 3) ± SD; SD—standard deviation. Mean values within rows with different letters are significantly different (*p* < 0.05). CB—wheat beer without lemongrass added; BL1—wheat beer with 1% *m*/*v* lemongrass added; BL2—wheat beer with 2.5% *m*/*v* lemongrass added; BL3—wheat beer with 5% *m*/*v* lemongrass added; GAE—equivalent of gallic acid; TE—expressed as trolox equivalent (mM TE/L).

**Table 3 molecules-27-09040-t003:** Microbiological stability of wheat beers enriched with lemongrass.

	Fermentation Day after Bottling of Wheat Beer	Number of Yeasts and Moulds [cfu mL^−1^]	Number of Mesophilic Aerobic Bacteria [cfu mL^−1^]	Number of Mesophilic Bacteria of Lactic Fermentation [cfu mL^−1^]
CB	15	9.3 ^cB.^ × 10⁴	5.7 ^bA^ × 10⁴	<1.0 ^aB^ × 10¹
	45	<1.0 ^aA^ × 10⁰	1.9 ^aA^ × 10³	<1.0 ^aA^ × 10⁰
BL1	15	8.4 ^cdB^ × 10⁵	1.1 ^dB^ × 10⁶	<1.0 ^aB^ × 10¹
	45	6.3 ^bA^ × 10²	4.7 ^bA^ × 10⁴	<1.0 ^aA^ × 10⁰
BL2	15	1.8 ^dB^ × 10⁶	6.8 ^cB^ × 10⁵	<1.0 ^aB^ × 10¹
	45	3.3 ^bA^ × 10²	8.5 ^bA^ × 10⁴	<1.0 ^aA^ × 10⁰
BL3	15	1.3 ^cB^ × 10⁵	6.4 ^bA^ × 10⁴	<1.0 ^aB^ × 10¹
	45	<1.0 ^aA^ × 10⁰	4.2 ^bA^ × 10⁴	<1.0 ^aA^ × 10⁰

Data are expressed as mean values (*n* = 3) ± SD; SD—standard deviation. The mean values within columns with different letters are significantly different (*p* < 0.05). Different small letters denote differences in the results between lemongrass concentrations in wheat beers, and different capital letters indicate differences between the measurements. CB—wheat beer without lemongrass added; BL1—wheat beer with 1% *m*/*v* lemongrass added; BL2—wheat beer with 2.5% *m*/*v* lemongrass added; BL3—wheat beer with 5% *m*/*v* lemongrass added.

**Table 4 molecules-27-09040-t004:** Sensory analysis of wheat beers fortified with lemongrass extract.

	CB	BL1	BL2	BL3
Aroma	4.11 ^a^ ± 0.74	4.73 ^a^ ± 0.79	4.45 ^a^ ± 0.69	4.54 ^a^ ± 0.69
Taste	3.34 ^a^ ± 0.82	3.73 ^a^ ± 1.10	4.27 ^c^ ± 0.65	4.09 ^b^ ± 0.83
Foam stability	3.52 ^a^ ± 0.53	3.64 ^ab^ ± 0.51	4.09 ^bc^ ± 0.85	4.18 ^c^ ± 0.87
Bitterness	3.47 ^a^ ± 0.52	3.04 ^a^ ± 0.89	3.64 ^a^ ± 1.03	3.18 ^a^ ± 0.75
Saturation	3.76 ^a^ ± 0.48	4.09 ^ab^ ± 0.83	4.45 ^b^ ± 0.69	4.09 ^ab^ ± 0.83
Overall impression	3.52 ^a^ ± 0.46	3.77 ^a^ ± 0.71	4.25 ^b^ ± 0.54	4.08 ^b^ ± 0.68

Data are expressed as mean values (*n* = 3) ± SD; SD—standard deviation. Mean values within rows with different letters are significantly different (*p* < 0.05). CB—wheat beer without lemongrass added; BL1—wheat beer with 1% *m*/*v* lemongrass added; BL2—wheat beer with 2.5% *m*/*v* lemongrass added; BL3—wheat beer with 5% *m*/*v* lemongrass added.

## Data Availability

Not applicable.

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
