# Peer review of "Effect of the Addition of Lemongrass (Cymbopogon citratus) on the Quality and Microbiological Stability of Craft Wheat Beers"

_molecules, 2022, doi:10.3390/molecules27249040_

Round 1

Reviewer 1 Report

The manuscript with the title "Effect of the addition of lemongrass (Cymbopogon citratus) on the quality and microbiological stability of craft wheat beers" addresses a trending topic. Innovative products with different finishes are up-to-date.

In this study, the effects of adding lemon gras to beer are analyzed. This seems to be a potential idea for a future product, where the aroma and antioxidant activity are improved. Challenges such as microbial stability and increased acidity are also presented.

The manuscript presents a new assortment of beer, with different additions of lemongrass. The analyzes are the basic ones. A medium level study.

The manuscript is properly organized, but I would have some recommendations.

In abstract: adding purpose, clearly presented.

In the introduction, a general presentation is made on the owners of lemongrass, but also of wheat beer.

The results are processed statistically and are properly presented.

Figure 1. I recommend that there be the same amount of beer in the glass. Relevant both for the appearance and for the foam

The materials used lack the characterization for lemongrass. How was it processed, preserved, in what form was it used?

I also recommend the technological scheme for beer production, and a percentage recipe. The description

"The raw material charge for wheat beer brewing consisted of 60% commercial barley malt and 40% wheat malt" - correctly it would be "craft beers with wheat"

Author Response

The authors are grateful for the contribution of the Reviewer.

According to the Reviewer's comments, the manuscript has been revised by a native speaker.

I also recommend the technological scheme for beer production, and a percentage recipe. The description

Answer:

The technological scheme was added (Figure 3.)

The materials used lack the characterization for lemongrass. How was it processed, preserved, in what form was it used?

Answer:

In manuscript was added: ' The purchased raw material was fresh, it was not 
processed in any way, it was packed in a vacuum package that was torn open immediately before using the lemongrass. The raw material was crushed by knife manually before being added to the wheat beers' (lines 316 - 318).

Figure 1. I recommend that there be the same amount of beer in the glass. Relevant both for the appearance and for the foam

Answer:

The same volume of the analyzed beer was poured into the glasses, but due to the properties of the beer enriched with lemongrass, the structure of the beer foam and its stability were varied, as can be seen in the pictures of beers attached in Figure 1.

"The raw material charge for wheat beer brewing consisted of 60% commercial barley malt and 40% wheat malt" - correctly it would be "craft beers with wheat"

Answer:

According to Kunze (2010), wheat beers usually contain from 40 to 60% wheat malt or unmalted wheat. [Kunze, W. Technology Brewing and Malting, 4th ed.; VLB Berlin: Berlin, Germany 2010; pp. 108, 843].

In abstract: adding purpose, clearly presented.

Answer:

In abstract was added purpose (lines 10-12).

Reviewer 2 Report

The MS entitle 'Effect of the addition of lemongrass (Cymbopogon citratus) on the quality and microbiological stability of craft wheat beers' describes the addition of bioactive compounds from lemongrass with enormous health promoting properties. This work could be used at practical grounds to make beers less harmful for consumption and microbiologically safe for longer storage time. Some minor changes are required as indicated in the attached PDF.

Author Response

The authors are grateful for the contribution of the Reviewer.

According to the reviewer's comments, the manuscript has been revised by a native speaker.

All comments in the PDF file have been corrected.

Round 2

Reviewer 1 Report

The authors responded to all my comments.